# Fertility Variation and Effective Population Size across Varying Acorn Yields in Turkey Oak (*Quercus cerris* L.): Implications for Seed Source Management

Nebi Bilir [1,†], Koeun Jeon [2,†], Ye-Ji Kim [2] and Kyu-Suk Kang [2,*]

1 Faculty of Forestry, Isparta University of Applied Sciences, 32260 Isparta, Turkey; nebibilir@isparta.edu.tr
2 Department of Agriculture, Forestry and Bioresources, Seoul National University, Seoul 08826, Republic of Korea; jke1014@snu.ac.kr (K.J.); kyeji1107@snu.ac.kr (Y.-J.K.)
* Correspondence: kangks84@snu.ac.kr
† These authors contributed equally to this work.

**Abstract:** This research examines the impact of varied acorn yields on the effective population size of Turkey oak (*Quercus cerris* L.) as assessed through the fertility averages of zygotic parents. We selected two distinct populations from the species' natural habitats based on their good and poor acorn production rates to investigate acorn production, growth attributes, and their interrelationships over three years of production and two years of growth data. Results showed that the population with good acorn production exhibited greater growth attributes and acorn yields compared to the poor acorn production population. Acorn production had lower coefficients of variation compared to growth attributes. Fertility variation in both populations was moderate, with a decrease in the effective number of parents from the population with abundant acorn production to the one with limited acorn production. The presence of mixed seeds from diverse populations had a detrimental impact on fertility variation and related metrics. Nonetheless, this study suggests that regions with limited acorn production still have the potential for natural regeneration due to their larger effective population size when coupled with appropriate forestry practices such as selective acorn harvesting to enhance genetic diversity. These findings emphasize the importance of accounting for fertility variation in the selection and management of seed sources, even within the context of a limited area and three years of data. Further research should be conducted in larger populations and over longer periods to draw more comprehensive conclusions.

**Keywords:** acorn production; growth attributes; genetic diversity; regeneration; forestry practices

## 1. Introduction

According to the latest forest inventory, Turkey has a total forest area of 23.3 million hectares, with 9.6 million ha being unproductive [1]. Oaks are the largest taxa, with 18 species and 11 sub-species [2] covering 6.8 million ha, of which 60% is unproductive [1]. Turkey oak (*Quercus cerris* L.) stands out as the predominant species in these oak populations due to its extensive natural distribution in Turkey and worldwide [3]. It can reach diameters of 1–1.20 m and heights of 30–35 m from sea level to 1500–1900 m with two sub-species, *Quercus cerris* var. *cerris* and *Q. c.* var. *austriaca* [2,3].

Turkey oak typically begins to produce acorns around the age of 60 years in natural stands [4]. Its monoecious, wind-pollinated flowers appear in April–May, resulting in the development of large, stalkless acorns measuring 2–3.5 (5) cm in length and 2 cm in width with densely covered with bristly acorn cups [5]. Although Turkey oak acorns take 18 months to mature, they exhibit abundant crops and readily germinate, making them suitable for both natural and artificial forestry practices [6]. The acorns are not only a key source for sustaining the species but also for culinary uses, being edible and used to sweeten food or mixed with flour for cake making [7]. Furthermore, Turkey oak acorns

serve as an important food source for wildlife. These fruits are known for their high tannin content, resulting in a strong astringent quality [8]. However, it is worth noting that good seed years occur only once every 3–4 years in this species [9]. Various biotic and abiotic factors can influence acorn/seed yield, though these effects have not yet been thoroughly studied in Turkey oak despite extensive research in other plant species, e.g., [10–15].

The species improves soil quality and provides sustenance for wildlife through its edible acorns. The species also has a natural distribution in countries such as France, Germany, Switzerland, Czechia, Slovakia, and Hungary [16]. It has been introduced to other European countries, Ukraine, North America, Argentina, and New Zealand [17]. Turkey oak is a deciduous tree native to southern Europe and Asia Minor, and it plays a dominant role in the mixed forests of the Mediterranean basin. It is categorized within section *Cerris*, a distinct section in the oak genus (*Quercus*) that includes Turkey oak [6].

Currently, Turkey has 13 seed stands covering 1216.2 hectares dedicated to various oak species. These include eight seed stands covering 259.6 hectares for Turkey oak, four seed stands covering 855.6 hectares for sessile oak (*Q. petraea* (Matt.) Liebl.), and one seed stand spanning 101 hectares for Kasnak oak (*Q. vulcanica* (Boiss. and Heldr. ex) Kotschy). However, it is important to note that, as of now, no seed orchards have been established for oaks [18]. Managing these seed sources is of great significance in the selection and establishment of seed/acorn production areas for forest owners, managers, and geneticists.

Fertility and linkage metrics are vital indicators that are easily accessible, cost-effective, and practical [19] for managing genetic resources and conservation areas, serving various purposes, i.e., [20–25]. Numerous theoretical and applied studies have investigated fertility data derived from strobili, seed, cone, acorn, and fruit production in both artificial and natural populations of diverse forest tree species [19,20,22,24,26–36]. However, in this study, we specifically focus on acorn production to examine fertility variation and effective population size in natural stands, also known as populations of Turkey oak.

The main purposes of this study are as follows: (1) to estimate the effect of growth characteristics on acorn production in natural seed stands of Turkey oak, (2) to compare acorn production between populations with good and poor acorn production, (3) to estimate the effective number of parents based on fertility variation in acorn production in natural stands of Turkey oak, and (4) to contribute to the establishment, selection, and management of current and future seed sources, as well as other forestry practices related to the species.

## 2. Materials and Methods

### 2.1. Stand and Tree Selections

The population with good acorn production (referred to as **GAP**, latitude 37°40′555″ N, longitude 30°31′941″ E, 930 m above sea level) had higher acorn production than the other population, while the population with comparatively poor acorn production (referred to as **PAP**, latitude 37°36′571″ N, longitude 30°52′840″ E, 960 m above sea level) (Figure 1) had lower acorn production. These populations were selected from their respective natural distribution areas, and they were approximately 8.16 km apart from each other. We sampled 100 trees with specific phenotypic characteristics, including symmetric crown diameter, larger diameters at the base and breast height, greater height, and straighter stems compared to others. A minimum distance of 100 m was maintained between each tree within the same population. Additional trees were sampled from the same populations in 2022 and 2023. Climatic data for the years 2021, 2022, and 2023 were obtained from the General Directorate of Meteorology and Climatology of Turkey [37] for the respective sites (Table 1). We used climatic data of a permanent local meteorological station (latitude 37°50′16″ N, longitude 30°52′19″ E, 940 m above sea level) in the description of the populations. The station was located at a distance of 17.3 km from the GAP and 25.8 km from the PAP populations.

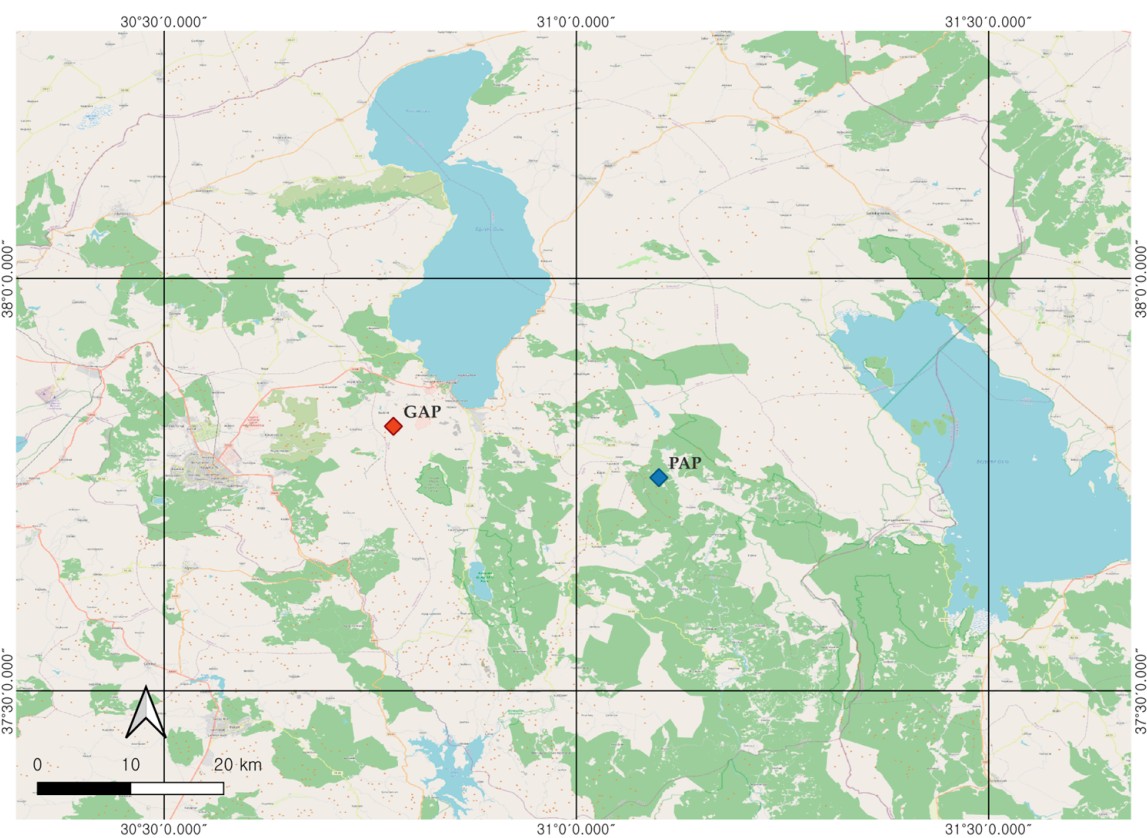

**Figure 1.** The location of the good acorn production population (**GAP**) and the poor acorn production population (**PAP**).

**Table 1.** Climatic characteristics (maximum temperature, minimum temperature, temperature averages, humidity averages, and total rainfall) for the years 2021, 2022, and 2023.

|  | Years/ Months | Jan | Feb | Mar | Apr | May | Jun | Jul | Aug | Sep | Oct | Nov | Dec |
|---|---|---|---|---|---|---|---|---|---|---|---|---|---|
| Maximum temperature (°C) | 21 | 15 | 17.5 | 17.6 | 28 | 33.3 | 31.9 | 35 | 37.5 | 31.7 | 24.6 | 23.9 | 14.5 |
|  | 22 | 14 | 12.3 | 16.4 | 26.7 | 30.1 | 29.7 | 33.8 | 34.2 | 31.6 | 31.4 | 20.8 | 14.6 |
|  | 23 | 14.5 | 18.9 | 18.6 | 22 | 25.5 | - | - | - | - | - | - | - |
| Minimum temperature (°C) | 21 | −4.7 | −4.0 | −1.1 | 0.8 | 6.4 | 9.2 | 15.1 | 12.9 | 7.5 | 3.6 | 0 | −5.0 |
|  | 22 | −9.9 | −5.4 | −5.5 | 2.5 | 5.2 | 13.5 | 12.9 | 0 | 6.6 | 2.1 | 0 | −0.4 |
|  | 23 | 2.4 | 7.4 | 0.3 | 2.9 | 4.1 | - | - | - | - | - | - | - |
| Average temperature (°C) | 21 | 5.6 | 6.2 | 6.8 | 12.9 | 19.2 | 19.9 | 25.7 | 25.5 | 19.8 | 14 | 10.8 | 6.2 |
|  | 22 | 2.1 | 3.5 | 3.4 | 14.2 | 16.8 | 21.9 | 24.7 | 25.1 | 20.5 | 15 | 10.2 | 7.6 |
|  | 23 | 5.1 | 3.3 | 9.0 | 11 | 15.7 | - | - | - | - | - | - | - |
| Average humidity (%) | 21 | 78.5 | 69.1 | 66.5 | 59.1 | 52.2 | 64.2 | 49.7 | 47.6 | 57.5 | 62.6 | 70.9 | 79.1 |
|  | 22 | 75.7 | 81.1 | 65.1 | 51.4 | 59.4 | 61.2 | 49.4 | 59.2 | 55.4 | 64.9 | 71.1 | 81.4 |
|  | 23 | 77.7 | 66.1 | 70.4 | 67.9 | 71.3 | - | - | - | - | - | - | - |
| Total rainfall (mm, kg/m²) | 21 | 182 | 50.6 | 60.8 | 10.0 | 7.0 | 90.0 | 6.4 | 0 | 15.8 | 15.6 | 64.4 | 185.8 |
|  | 22 | 168.8 | 322.4 | 127.8 | 23.2 | 17.8 | 14.8 | 3.8 | 18.8 | 24.2 | 8.8 | 43.2 | 72.0 |
|  | 23 | 114.0 | 7.6 | 170.8 | 131.6 | 59.6 | - | - | - | - | - | - | - |

### 2.2. Data Collection and Analysis

Data on tree height (**H**), diameter at the base ($D_0$), diameter at breast height ($D_{1.30}$), and crown diameter (**CD**) were collected at the end of the growing period in 2022 and in the middle of the growth period (mid-June) in 2023. Tree heights were measured using a Vertex IV hypsometer (Haglöf, Järfälla, Sweden), a widely used tool in forestry practices.

Additionally, the diameter at the base and the diameter at breast height of each tree were measured with a caliper. Averages of crown diameters in the east–west and north–south directions were measured using a tape measure due to the species' asymmetrical crown structure. The number of mature acorns ($AN_2$), which were 2 years old, was counted at the end of the growing period in 2022. The numbers of one-year-old acorns ($AN_1$) and acorns that were already pollinated ($AN_0$) were counted, and growth characteristics were measured (Figure 2). The species has a flowering and fruiting cycle of approximately 18 months [38]; $AN_2$ in 2022 and $AN_1$ and $AN_0$ in 2023 can be considered indicators of the acorn production for 2022, 2023, and 2024, respectively, in the future. However, various factors could affect the maturation periods of acorns.

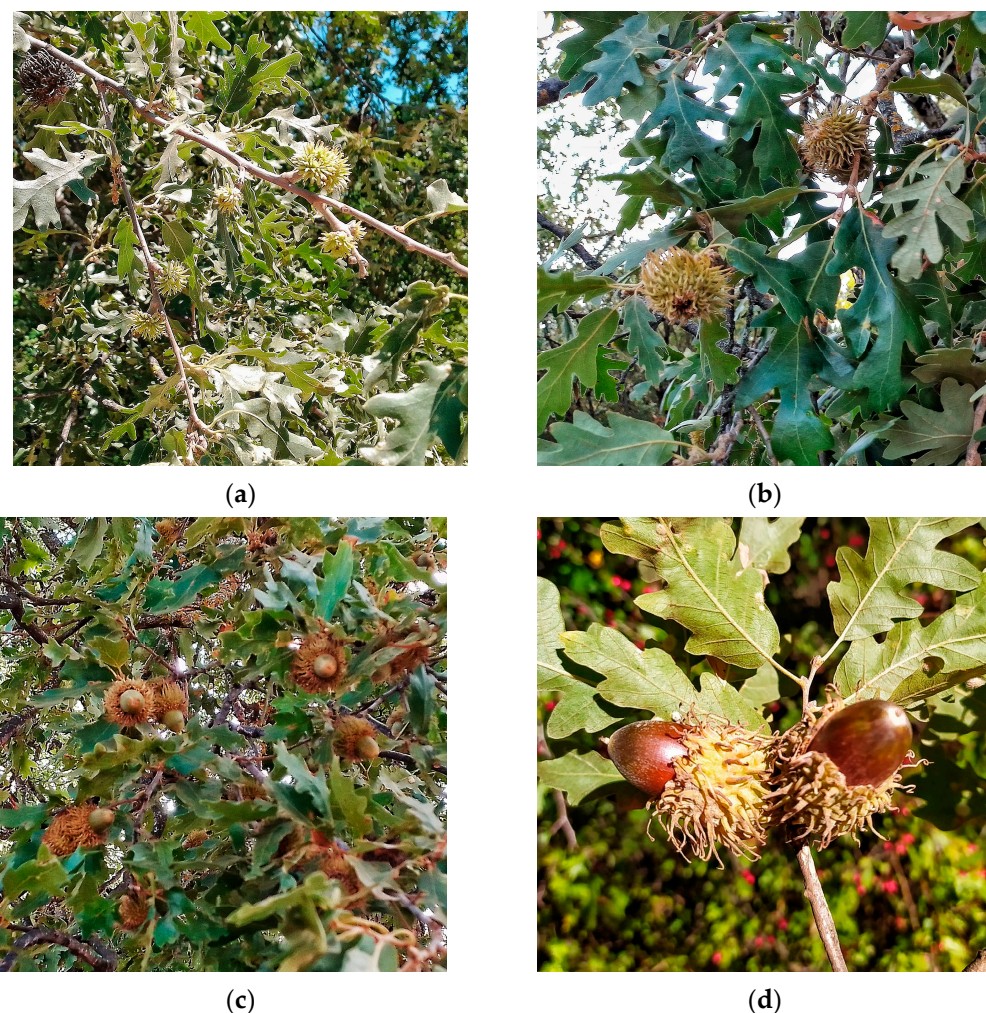

(**a**)

(**b**)

(**c**)

(**d**)

**Figure 2.** Acorns that have already been pollinated ($AN_0$) (**a**), one-year-old acorns ($AN_1$) (**b**), and mature acorns that are two years old ($AN_2$) (**c**,**d**).

To compare the populations in terms of acorn production and growth characteristics, the following linear model of analysis of variance (ANOVA) was performed using the SAS package [39].

$$Y_{ij} = \mu + C_i + e_{ij} \tag{1}$$

where $Y_{ij}$ is the characteristic from the $j$th tree of the $i$th population, $\mu$ is the overall mean, $C_i$ is the random effect of the $i$th population, and $e_{ij}$ is the random error.

Phenotypic Pearson correlations among acorn production and growth characteristics were estimated for each population using the package.

*2.3. Fertility Variation and Effective Population Size*

The variation in acorn fertility ($\Psi$) was estimated following Kang and Lindgren [28] as:

$$\Psi = N\Sigma_{i=1}^{N} Acorn_i^2 = CV_{Acorn}^2 + 1 \quad (2)$$

where $N$ is the census number or the number of evaluated trees; $Acorn_i$ is the acorn fertility, also referred to as total fertility, for the $i$th individual; and $CV_{Acorn}$ is the coefficient of variation in total fertility. In this paper, acorn fertility represents the total contribution of zygotic parents.

The effective number of parents ($N_p$) was estimated based on acorn fertility ($\Psi$) and census number ($N$), as described by Kang et al. [21]:

$$N_p = N/\Psi \quad (3)$$

The relative effective numbers of parents ($N_r$) were estimated by comparing the effective number of parents ($N_p$) to the census number ($N$):

$$N_r = N_p/N \quad (4)$$

Gene diversity (GD) was estimated based on the effective number of parents ($N_p$) according to Kang and Lindgren [26]:

$$GD = 1 - 0.5/N_p \quad (5)$$

*2.4. Parental Balance Curve*

The contribution of each tree in terms of uniformity and homogeneity was analyzed using a parental balance curve. The curve was plotted by sorting families in descending order on the *x*-axis while representing the accumulated contribution of each family to the total flower production on the *y*-axis. This curve illustrates the gamete contribution of specific families or clones using a cumulative contribution curve [40]. Therefore, this cumulative contribution curve is crucial for improving the expectation of equal tree contributions in the establishment and management of seed sources in the future [19,36,41].

## 3. Results and Discussion

### 3.1. Acorn Production, Growth Characteristics, and Their Relations

The averages of the growth characteristics were significantly higher in the population with good acorn yield compared to the population with poor acorn yield (Table 2). The good acorn yield population ($AN_0$ = 407, $AN_1$ = 158, and $AN_2$ = 803) exhibited higher acorn production than the poor acorn yield population ($AN_0$ = 170, $AN_1$ = 85, and $AN_2$ = 254) for all acorn periods. The average acorn production over the pooled years was 456 in the good acorn yield population (GAP) and 170 in the poor acorn yield population (PAP) (Table 2). Statistically significant differences ($p < 0.01$) were observed between populations and years within populations for acorn production according to the results of the analysis of variance. Individual trees within each population displayed differences in these characteristics, with differences exceeding 16 times for mature acorn production in both populations (Table 2). The top ten productive individual trees accounted for 23.1% of the total acorns in the GAP and 27.7% in the PAP groups. The coefficients of variation were higher in the PAP group compared to the GAP group for all years (85.1% in GAP, 83.1% in PAP). The highest variation occurred in the highest acorn yield for both populations (Table 2). These results emphasize the importance of considering the acorn collection year for this species.

**Table 2.** Averages ($\overline{X}$), ranges, and coefficients of variation (CV) for acorn production within the populations. GAP refers to the population with good acorn production, while PAP represents the population with poor acorn production. The acorns are categorized as $AN_2$ (two-year-old mature acorns), $AN_1$ (one-year-old acorns), and $AN_0$ (pollinated acorns in that particular year).

|  |  | $AN_2$ | $AN_1$ | $AN_0$ | Total |
|---|---|---|---|---|---|
| **GAP** | $\overline{X}$ | 803 | 158 | 407 | 456 |
|  | Ranges | 160–2700 | 72–374 | 164–812 | 72–2700 |
|  | CV% | 58.1 | 38.9 | 34.3 | 85.1 |
| **PAP** | $\overline{X}$ | 254 | 85 | 70 | 170 |
|  | Ranges | 54–880 | 32–270 | 26–426 | 26–880 |
|  | CV% | 73.2 | 45.1 | 58.1 | 83.1 |

Populations and individual trees within populations displayed significant variations in mature and immature acorn production (Table 2). In a clonal seed orchard of Sawtooth oak (*Quercus acutissima* (Qac)), the clonal average of acorn production varied between 5.2 and 327.5 over eight years, with approximately 60 out of 94 clones consistently producing acorns almost every year [31]. The *Quercus* genus comprises approximately 531 species worldwide, distributed across the Americas, Asia, Malaysia, Europe, and North Africa [42]. Variations among species and taxa are expected, as significant differences in reproductive characteristics have been reported within populations and among populations in natural stands, e.g., [34], and seed orchards, e.g., [22,35,43] of different forest tree species, including Oaks such as northern red oak (*Quercus rubra* L.) and black oak (*Q. velutina* Lam.) [44] and white oak (*Q. alba* L.) [45]. These variations may be influenced by genetic and environmental factors [10,46].

Previous studies have reported that acorn production in oaks can be affected by soil characteristics, climatic factors during the vegetation period, genetic structure of individual trees [47], or stand structures [48]. However, in considering the pollination of $AN_2$, $AN_1$, and $AN_0$ in 2021, 2022, and 2023, respectively, $AN_1$ exhibited the lowest acorn yield in both the GAP and the PAP populations (Table 2). This could be related to the heavy rainfall in February 2022, as depicted in Table 1, which potentially led to pre-dispersal failure [49]. In scenarios involving wind pollination, such as oaks, intense precipitation can contribute to nectar dilution, pollen deterioration, and the dissipation of volatiles [50]. Furthermore, the potential for predicting weather influences using AI (artificial intelligence), such as machine learning, has been suggested [51].

The growth characteristics exhibited distinct patterns between the two populations, the good acorn year population (GAP) and the poor acorn year population (PAP), as outlined in Table 3. Results of the analysis of variance revealed highly significant differences ($p < 0.01$) in growth characteristics across years between the populations. Notably, the impact of growth characteristics on acorn production was significant ($p < 0.05$), as evidenced by the correlation analysis conducted in both the GAP and the PAP populations (Table 4). Significant ($p < 0.05$) correlations were estimated among the acorn production and the growth characteristics ($r > 0.521$ for $AN_0$; $r > 0.452$ for $AN_1$; $r > 0.482$ for $AN_2$) in pooled populations. Additionally, positive and significant ($p < 0.05$) relationships among growth characteristics were observed in each population and their combination for the years. The interrelation between $AN_1$ and $AN_0$ within each population was also positively significant ($p < 0.05$) (Table 4). Plants allocate their energy for growth or reproductive purposes in different years [20].

**Table 3.** Averages ($\overline{X}$), ranges, and coefficients of variation (CV) for growth characteristics within population. The populations are denoted as GAP (good acorn production population) and PAP (poor acorn production population). In this context, H refers to tree height, and $D_0$, $D_{1.3}$, and CD represent diameter at the base, diameter at breast height, and crown diameter, respectively.

| | | | H (m) | $D_0$ (cm) | $D_{1.30}$ (cm) | CD (m) |
|---|---|---|---|---|---|---|
| **2022** | **GAP** | $\overline{X}$ | 18.2 | 93.3 | 73.3 | 14.5 |
| | | **Ranges** | 12.3–24.5 | 58.0–145.0 | 40.0–115.0 | 8.7–22.8 |
| | | **CV%** | 12.7 | 21.9 | 24 | 21.6 |
| | **PAP** | $\overline{X}$ | 16.5 | 70.4 | 52.4 | 11.7 |
| | | **Ranges** | 10.2–21.5 | 35.0–170.0 | 28.0–100.0 | 5.5–17.2 |
| | | **CV%** | 13 | 24.2 | 23.8 | 23.3 |
| **2023** | **GAP** | $\overline{X}$ | 20.1 | 93.7 | 75.3 | 14.6 |
| | | **Ranges** | 12.5–25.2 | 60.0–132.0 | 43.0–118.0 | 9.5–18.7 |
| | | **CV%** | 12.5 | 16.8 | 22.5 | 13.9 |
| | **PAP** | $\overline{X}$ | 17.6 | 73.3 | 56.4 | 12.4 |
| | | **Ranges** | 7.7–25.1 | 37.0–135.0 | 28.0–102.0 | 7.7–19.5 |
| | | **CV%** | 18.7 | 24 | 27.3 | 21.8 |

**Table 4.** Relationships among characteristics in the two populations: GAP (good acorn production population, above the diagonal) and PAP (poor acorn production population, below the diagonal).

| | r | $AN_2$ | H | $D_0$ | $D_{1.30}$ | | CD |
|---|---|---|---|---|---|---|---|
| **2022** | $AN_2$ | - | 0.385 ** | 0.431 ** | 0.416 ** | | 0.486 ** |
| | H | 0.389 ** | - | 0.575 ** | 0.613 ** | | 0.638 ** |
| | $D_0$ | 0.402 ** | 0.433 ** | - | 0.906 ** | | 0.570 ** |
| | $D_{1.30}$ | 0.556 ** | 0.546 ** | 0.799 ** | - | | 0.651 ** |
| | CD | 0.296 ** | 0.545 ** | 0.389 ** | 0.532 ** | | - |

| | r | $AN_1$ | $AN_0$ | H | $D_0$ | $D_{1.30}$ | CD |
|---|---|---|---|---|---|---|---|
| **2023** | $AN_1$ | - | 0.658 ** | 0.327 ** | 0.453 ** | 0.436 ** | 0.405 ** |
| | $AN_0$ | 0.362 ** | - | 0.368 ** | 0.381 ** | 0.389 ** | 0.400 ** |
| | H | 0.325 ** | 0.474 ** | - | 0.384 ** | 0.398 ** | 0.432 ** |
| | $D_0$ | 0.492 ** | 0.466 ** | 0.612 ** | - | 0.936 ** | 0.612 ** |
| | $D_{1.30}$ | 0.445 ** | 0.498 ** | 0.583 ** | 0.901 ** | - | 0.604 ** |
| | CD | 0.356 ** | 0.514 ** | 0.623 ** | 0.701 ** | 0.702 ** | - |

**: Correlation is significant at the 0.01 level.

A positive correlation was observed between tree size and acorn production in Mongolian oak (*Quercus mongolica* Fisch.) by Noh et al. [11]. Similarly, another investigation by Stăncioiu et al. [12] suggested that the diameter of the crown effectively reflects the health of Turkey oak trees. Moreover, Kim et al. [13] found a positive impact of crown diameter and tree age on acorn production in a seed orchard of *Quercus acutissima*. When examining tamarind (*Tamarindus indica* L.) and neem (*Azadirachta indica* A. Juss), Zhang et al. [24] found that tree height and stem girth significantly influenced both male and female flowering in seed stands. However, a different scenario emerged for loblolly pine (*Pinus taeda* L.) [14] and scots pine (*P. sylvestris* L.) [15], with negative genetic correlations between flowering and growth traits. Contrarily, lodgepole pine (*Pinus contorta* (Dougl.)) [52] displayed low genetic correlations between tree height and flowering. These findings have significant implications for the selection, establishment, and management of seed sources, including practices like pruning and spacing.

Acorn production ($AN_2$) was best predicted by diameter at breast height ($D_{1.30}$) based on regression analysis for the pooled populations (Figure 3a) and the poor acorn year population (Figure 3b). However, for the good seed year population (Figure 3c), diameter at the base ($D_0$) was identified as a better predictor. In the Brazilian Atlantic forest, diameter at breast height ($D_{1.30}$) appeared to be a reasonable predictor for the number of flower [43].

When considering oak species in eastern United States, the production of acorns increased as the $D_{1.30}$ of the tree expanded, suggesting that an augmentation in $D_{1.30}$ aligns with a rise in crown diameter [53]. Additionally, low and positive relationships between diameter at breast height, crown diameter, and acorn production were also found [54–56]. These results can be used in future predictive models of acorn yield.

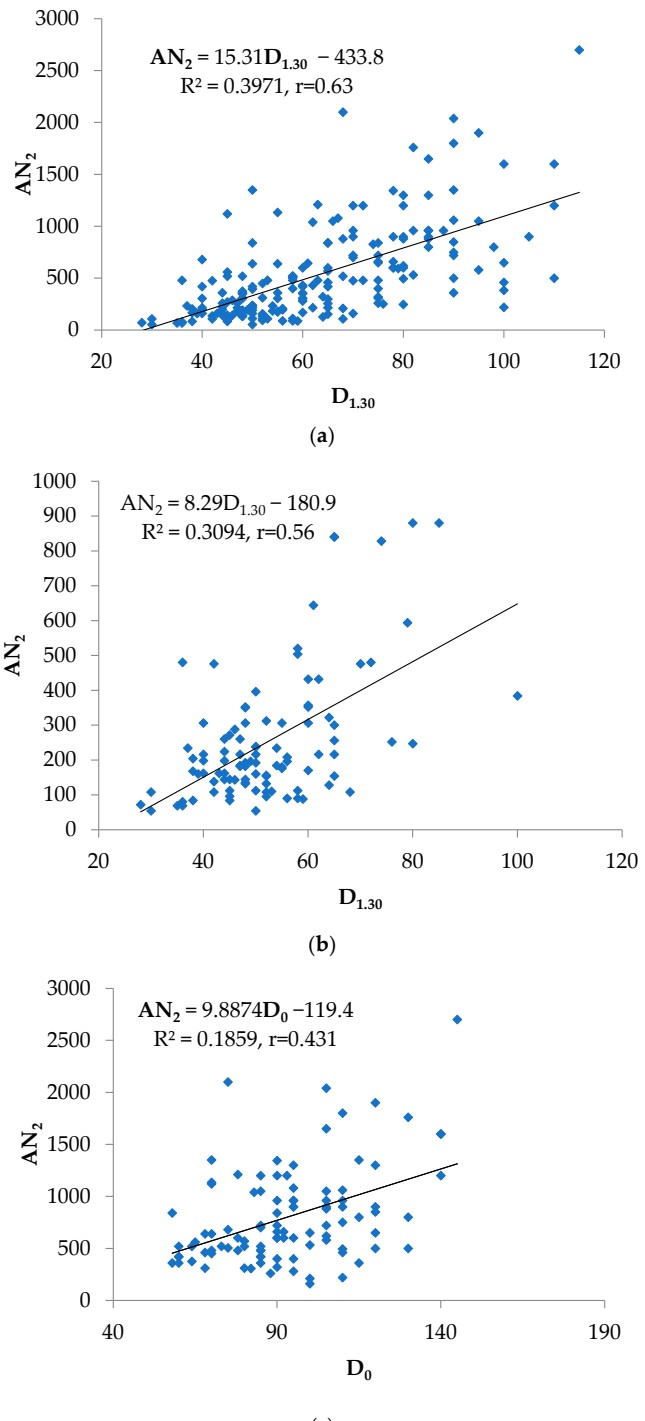

**Figure 3.** Regression analyses of acorn production ($AN_2$) and diameter at breast height ($D_{1.30}$) in pooled populations (**a**) and the poor acorn year population (**b**) and of acorn production ($AN_2$) and diameter at the base ($D_0$) in the good seed year population (**c**).

### 3.2. Fertility Variation, Effective Number of Parents, Gene Diversity, and Parental Balance

The cumulative contribution of the trees to the overall fertility estimates for acorn production in each population and the pooled populations is shown in Figure 4. The parental balance curve (Figure 4) can serve as a guide for balancing contributions, especially in the selection of individual trees. The figure demonstrates that the GAP population exhibited a much closer to equal contribution in acorn production compared to the PAP population, while the pooled populations had a more homogenous contribution in $AN_1$ and $AN_2$ than in $AN_0$ (Figure 4). However, it should be noted that the curves of $AN_0$ and $AN_1$ may change during maturation.

The acorn fertility, representing the collective contribution of zygotic parents (i.e., total fertility), exhibited a moderate trend for both populations, as indicated in Table 3. However, fertility variation among individual trees proved narrower in the GAP group ($\Psi = 1.33$, CV% = 58) than in the PAP group ($\Psi = 1.53$, CV% = 73), as presented in Table 5. This pattern correlates with the parental balance curves used to characterize highly or poorly productive individual trees based on cumulative gamete contribution, as depicted in Figure 4. With a coefficient of variation (CV) up to 140% ($\Psi \leq 3$), this level of fertility variation was considered acceptable as a generalized heuristic rule for natural populations, a concept previously outlined by Lindgren and Prescher [25].

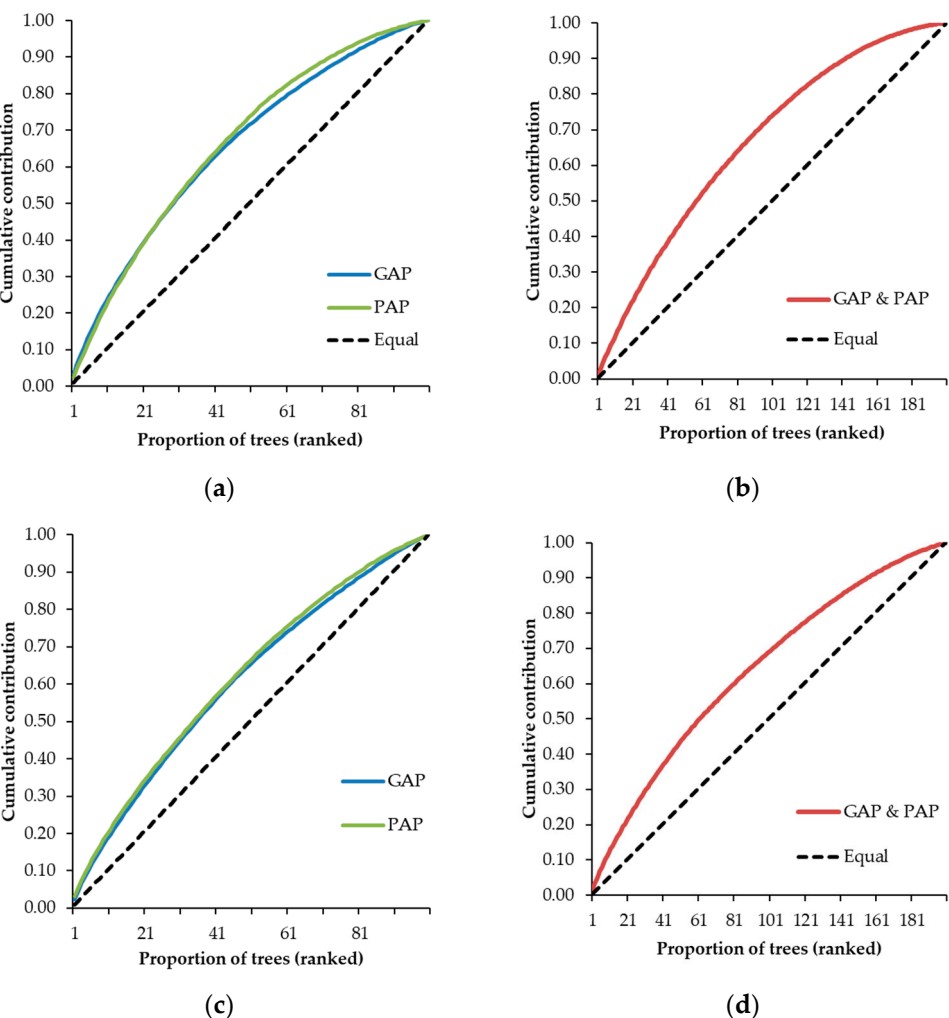

**Figure 4.** *Cont.*

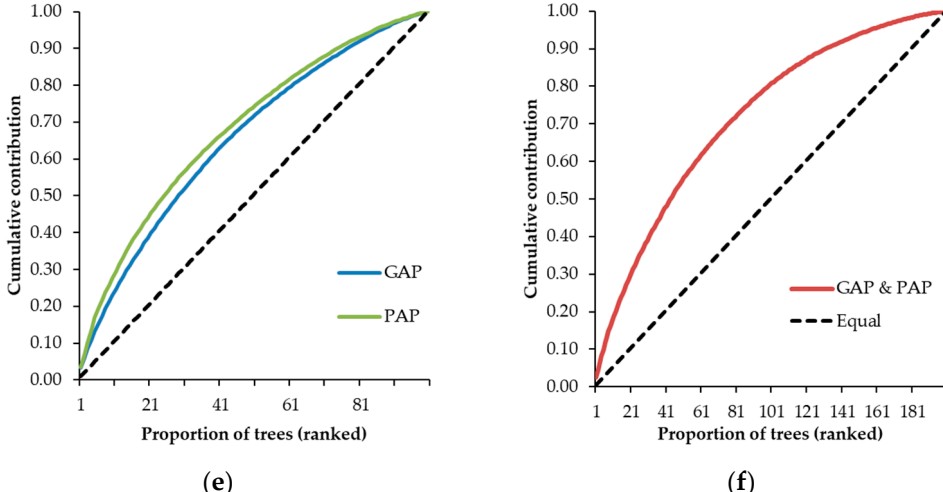

**(e)**                    **(f)**

**Figure 4.** Parental balance curves for two different populations: GAP (good acorn production) and PAP (poor acorn production), as well as the pooled population. The curves labeled (**a**,**b**) represent the data for $AN_2$, (**c**,**d**) for $AN_1$, and (**e**,**f**) for $AN_0$. Curves (**a**,**c**,**e**) correspond to the two populations (GAP and PAP), while curves (**b**,**d**,**f**) pertain to the pooled population.

**Table 5.** Acorn fertility ($\Psi$), the effective number of parents ($N_p$), the relative effective numbers of parents ($N_r$), and gene diversity (GD) for the populations: the good acorn production population (GAP), the poor acorn production population (PAP), and the pooled population.

|  |  | $\Psi$ | $N_p$ | $N_r$ (%) | GD |
|---|---|---|---|---|---|
| | $AN_2$ | 1.33 | 74.9 | 75 | 0.993 |
| **GAP** | $AN_1$ | 1.15 | 86.9 | 87 | 0.994 |
| | $AN_0$ | 1.12 | 89.6 | 90 | 0.994 |
| | $AN_2$ | 1.53 | 65.3 | 65 | 0.992 |
| **PAP** | $AN_1$ | 1.2 | 83.3 | 83 | 0.994 |
| | $AN_0$ | 1.33 | 74.9 | 75 | 0.993 |
| | $AN_2$ | 1.72 | 116.4 | 58 | 0.996 |
| **Total** | $AN_1$ | 1.27 | 158 | 79 | 0.997 |
| | $AN_0$ | 1.34 | 148.9 | 74 | 0.997 |

The findings suggest that the suboptimal PAP could be utilized for natural regeneration and other forestry practices due to its diverse attributes [57]. Furthermore, the results highlight the significance of equitable gamete contribution in minimizing inter-individual variation within the species or area rather than focusing solely on increased acorn production. However, fertility variation and effective numbers may fluctuate over time in seed stands of *Tamarindus indica* and *Azadirachta indica* [24], underscoring the need for continuous monitoring and estimation of these indicators.

The effective number of parents, which reflects $\Psi$, decreased from 74.9 (constituting 75% of the census number) in the GAP population to 65.3 (65%) in the PAP population, while in the pooled populations, it stood at 116.4% [58]. These observations highlight that mixing populations/acorns negatively impacted enhancing fertility variation and linkage parameters, contrary to suggestions in various forest tree species, e.g., [10,12,54]. However, this impact could be balanced by traditional or genetic forest management in seed sources or areas of natural regeneration to transmit the present gene diversity to subsequent generations [59].

The discrepancy in estimated gene diversity was 0.001 between the GAP (0.9933) and the PAP (0.9923) populations, but it increased by 0.0024 in pooled populations (0.9957), as detailed in Table 5. Kang et al. [31] previously reported that the effective population size

($N_p$) was higher in moderate and good acorn production years within a clonal seed orchard of *Quercus acutissima*. However, it is worth noting that the present study encompassed data collected over a three-year span solely within natural stands. Given the variation in acorn production among diverse *Quercus* species Op. cit., [44], an extended data collection period would offer a more comprehensive viewpoint. Additionally, incorporating molecular analysis to ascertain the genetic distance between individual trees in these stands would be prudent. Such an analysis could provide insights into genetic relatedness among trees, forming a foundational resource for the future organization of seed orchards in the selection of plus trees, aligning with fitness theory [60].

## 4. Conclusions

Positive and significant correlations between growth attributes and acorn production underscored their importance in strategic selection, formation, and management of seed sources (i.e., spacing, pruning, other tending practices).

Furthermore, fertility variation within individual trees was found to be moderate in both populations, with a narrower range observed in the good acorn production group. This suggests the potential utilization of populations with poor acorn yields for natural regeneration and seed collection practices. However, it also became evident that the amalgamation of seeds from diverse populations had a detrimental effect on fertility variation and linked parameters such as low gene diversity in the seed crop. Nevertheless, this negative effect can be addressed through traditional or genetic forest management practices aimed at preserving gene diversity in seed sources and natural regeneration areas.

It is worth noting that this study is limited by its two-year data collection period (2022 and 2023), which highlights the need for extended investigations involving larger sample sizes and longer observation periods to provide a more comprehensive understanding. The insights presented in this research contribute valuable knowledge for the consideration, establishment, and supervision of Turkey oak seed sources. Yet, further research remains essential to build upon these findings and enhance our comprehension of the species's reproductive dynamics and genetic variability.

**Author Contributions:** Conceptualization, N.B.; data curation, N.B. and K.J.; investigation, N.B.; methodology, N.B. and K.J.; project administration, N.B. and K.-S.K.; resources, N.B.; software, N.B.; supervision, K.-S.K.; validation, N.B., K.J., K.-S.K. and Y.-J.K.; visualization, K.J.; writing—original draft, N.B.; writing—review and editing, K.J. All authors have read and agreed to the published version of the manuscript.

**Funding:** This study was carried out with the support of the R&D Program for Forest Science Technology (Project No. FTIS 2022458B10-2224-0201) provided by the Korea Forest Service (Korea Forestry Promotion Institute).

**Data Availability Statement:** The data presented in this study are available on request from the corresponding author.

**Acknowledgments:** The authors thank the regional forest directorate for their administrative support. We also thank the anonymous reviewers who made valuable comments which helped to improve the manuscript.

**Conflicts of Interest:** The authors declare no conflict of interest.

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
