# Peer review of "Fertility Variation and Effective Population Size across Varying Acorn Yields in Turkey Oak (Quercus cerris L.): Implications for Seed Source Management"

_forests, doi:10.3390/f14112222_

Round 1
Reviewer 1 Report
Comments and Suggestions for Authors
please see the attachment

Author Response
Dear Editorial Office of Forests and Reviewer,
Thank you very much for taking the time to review this manuscript (ID: forests- 2630112). Please find the detailed responses below and the corresponding revisions highlighted in track changes based on Reviewers comment.
Comments 1: Lines 74-75 – methods describing how “good” and “poor” acorn production areas needs to be delineated here.
Response 1: We appreciate your observation, and we concur with the reviewer's feedback. Additional details have been incorporated in response to this suggestion. It's worth noting that there are no specific scales or numerical criteria in place for GAP and PAP. In the revised version, we've placed greater emphasis on distinguishing the annual data as either higher or lower.
Comments 2: Lines 77-78 – description of crown characterization should be included here.
Response 2: We agree with the reviewer. The content has added to the sentence.
Comment 3: Lines 90-93 – many things can happen to two-year acorns from onset of pollination to maturation. Numbers for AN1 and AN0 could drastically change with the onset of drought, pest problems, and late winter freezes. Conversely, the AN2 crop may be of acorns which are leftovers from past pest and environmental events.
Response 3: We agree with the reviewer. The sentence was softened based on the comment.
Comment 4: Lines 100-110 – Kang’s work was with Pinus, which has a different reproductive system that Quercus.
Response 4: Thank you for pointing this out. Considering Kang's research, specifically his studies on "Clonal Variation in Acorn Production and its Influence on Effective Population Size in a Quercus acutissima Seed Orchard" from 2010, we have incorporated this reference to establish a link between Kang's research and the Quercus genus. We appreciate your bringing this matter to our attention. Thank you for bringing this to our attention.
Comment 5: Line 112 – the authors use “family,” which implies genetically relatedness. However, there is nothing presented in the Materials and Methods that suggests the authors performed some kind of biochemical test to confirm relatedness.
Response 5: We thank to the reviewer for the comment. The authors agree to this. It has been changed from “family” to “individual”.
Comment 6: Lines 246-247 – the study was not a three-year study, but a one-year study of different acorn development on selected trees.
Response 6: We thank to the reviewer for the comment. Nevertheless, we have chosen to utilize a three-year dataset, comprising two years of acorn production and one year of pollinated flower data.
Comment 7: Quality of English Language
Response 7: The authors have the manuscript checked by a colleague fluent in English writing.
- General comments
We thank to the reviewer thorough review of our manuscript. We understand from the concerns regarding the extrapolation of one year of data on Turkey Oak acorn development to represent three years. To clarify, our dataset comprises two years of acorn yield data, encompassing 2021 and 2022, in addition to one year of pollinated flower data in 2023. We acknowledge the potential impact of environmental and pest factors on acorn production and will address this issue in the revised manuscript. We also commit to refining our methodology for characterizing Turkey Oak populations as 'good' or 'poor' acorn producers, considering regional variations.
The reviewer’s suggestion to measure crown area instead of crown diameter has been duly noted. While we recognize the merits with the suggestion, we opted for crown diameter measurements as they offer practicality and ease of field application.
We thank for recognition of the promise in our observations concerning the various stages of acorn development and their potential to elucidate the relationship between pollinated flowers and second-year acorns. We will incorporate this insight into the revised version of our work.
We thank to the reviewer for their valuable comments in improving the manuscript.
Sincerely,
Kyu-Suk Kang, Professor
Department of Agriculture, Forestry and Bioresources
College of Agriculture and Life Sciences
Seoul National University, Seoul 08826
Republic of Korea
Reviewer 2 Report
Comments and Suggestions for Authors
In the present work an evaluation of the influence of varied acorn yields on the effective population size of Turkey oak (Quercus cerris L.) was performed in two natural populations with. The populations were selected based on their good and poor acorn production rates.
Introduction
Line 30-32: The authors should be include an article in wich are mentionated these aspects, to support the web page information.
Line 45: “…..Quercus which includes species [6]”. The authors meant "other species"?
Line 49: “..four seed stands covering 259.6 ha of Turkey oak,…”; In this paragraph the author refers to Quercus cerris?; this should be clear.
Materials and Methods
How the authors establish the GAP and PAP oaks populations. The number of acorns produced by tree was estimated in previous years? Please, give more information on how these populations were selected.
How many tree approximately include each population?
The acorn fertility (Ψ) calculations is not clear, the author could be a better explanation of each components of the equation. N, census numbers, is the number of evaluated tree?
In the Parental balance curve, the families are all acorn of one tree?
Results
Line 120 “The averages growth characteristics were higher in the good acorn yield population..”. In this paragraph, are presented the production of acorns, it is recommended that change the averages growth by the numbers of acorns production.
Line 202: Please include what mean the sigla DBH.
Line 205. In figure caption the authors should include the description of each figure (a, b and c) to avoid confusion.
Author Response
Dear Editorial Office of Forests and Reviewer,
Thank you very much for taking the time to review this manuscript (ID: forests- 2630112). Please find the detailed responses below and the corresponding revisions highlighted in track changes in the re-submitted files.
Comment 1: Line 30-32: The authors should be include an article in wich are mentionated these aspects, to support the web page information.
Response 1: We agree with the reviewer. It could be better. However, forest inventory of Turkey was announced annually as on-line at website of Directorate General of Forestry of Turkey without published material.
Comment 2: Line 45: “…..Quercus which includes species [6]”. The authors meant "other species"?
Response 2: We agree with the reviewer. The sentence in line 45 was clarified.
Comment 3: Line 49: “..four seed stands covering 259.6 ha of Turkey oak,…”; In this paragraph the author refers to Quercus cerris?; this should be clear.
Response 3: We agree with the reviewer. The sentence in line 49 was clarified.
Comment 4: How the authors establish the GAP and PAP oaks populations. The number of acorns produced by tree was estimated in previous years? Please, give more information on how these populations were selected.
Response 4: We agree with the reviewer. It was explained as data collection year.
Comment 5: How many tree approximately include each population?
Response 5: “One hundred trees” was changed to “100 trees” in line 77 of subtitle 2.1 to be noticed easily for readers.
Comment 6: The acorn fertility (Ψ) calculations is not clear, the author could be a better explanation of each components of the equation. N, census numbers, is the number of evaluated tree?
Response 6: The components were tried to better explanation in revised version.
Comment 7: In the Parental balance curve, the families are all acorn of one tree?
Response 7: We are sorry for the confusion. We changed the family into individual.
Comment 8: Line 120 “The averages growth characteristics were higher in the good acorn yield population..”. In this paragraph, are presented the production of acorns, it is recommended that change the averages growth by the numbers of acorns production.
Response 8: We agree with the reviewer. The sentence in line 120 was improved.
Comment 9: Line 202: Please include what mean the sigla DBH.
Response 9: We agree with the reviewer. DBH was given as “diameter at breast height” in revised version.
Comment 10: Line 205. In figure caption the authors should include the description of each figure (a, b and c) to avoid confusion.
Response 10: We agree with the reviewer. The caption was improved to avoid confusion.
We thank to the reviewer for their valuable comments in improving the manuscript.
Sincerely,
Kyu-Suk Kang, Professor
Department of Agriculture, Forestry and Bioresources
College of Agriculture and Life Sciences
Seoul National University, Seoul 08826
Republic of Korea
Reviewer 3 Report
Comments and Suggestions for Authors
The present study investigates the relationships between acorn yields, growth characteristics, and fertility variation, which have significant implications for forest management. Overall, this is a fulfilling research paper that provides valuable insights on the forest management. However, some questions should be solved before publication. Therefore, I suggest that this paper is published after revising.
The major revisions:
1) Page 1, Line 29, about introduction, The focus of this study is seed yield; however, the introduction does not address certain pertinent studies on seed yield.
2) Page 2, Line 74-75, “A population with good acorn production (GAP, lat. 37°40'555'' N, long. 30°31'941'' E, 930 m asl.) and another population with poor acorn production...”. What are the definitions and criteria for GAP and PAP? Is the current year's yield or the long-term acorn production used as a reference?
3) Page 2, Line 76-77, “based on criteria”. What is the criteria? The standard should specify specific values or ranges instead of using descriptive terms such as larger diameter, greater height, and straighter stem.
4) Page 3, Line 79-80, “The sampled trees were renewed each year at the same sites”, What is the meaning of this sentence? Specifically, what does “renewed” ? Changing sample trees annually according to seed production? If so, how was the three-year experiment conducted?
5) Page 3, Line 82-83, “Climatic characteristics...”,The location of this climatological site is not specified. The distance between the GAP and PAP sites has not been provided. The presentation of climate characteristics differences between two sites should be considered. Based on your map, it can be inferred that GAP site is situated in closer proximity to the Black Sea, whereas PAP site appears to be more inland. Does the variation in microclimate, particularly humidity account for the disparity in seed yield? Therefore, it is necessary to consider the impact of climate on the outcomes of this study.
6) Page 3, Line 85-86, “Tree height (H), diameter at base (D0), diameter at breast height (D1.30) and crown 85 diameter (CD)”, The process of measuring these indicators should be described in detail, including the specific method used, the tools employed, and the number of samples taken.
7) Page 3, Line 90-93, “...AN2 in 2022, and AN1 and AN0 in 2023 could represent the acorn production of 2022, 2023, and 2024, respectively, in the future.”. Does it accurately reflect the potential of future production? The fact is that a significant proportion of pollinated seeds fail to reach maturity and achieve seed yield due to various factors, including drought.
8) Page 4, Line 96-99, “The one-way analysis of variance (ANOVA) was performed to compare the populations in terms of acorn production and growth characteristics. Phenotypic Pearson correlations among acorn production and growth characteristics were estimated within the populations by SAS package”. The description of the data analysis is rather rudimentary, lacking clear elucidation on the analysis process and parameters. Additionally, the depiction of the analysis results is also quite ambiguous in the result, rendering it difficult to comprehend easily.
9) Page 5, Line 120, “3.1. Acorn production, growth characteristics and their relations”. What was the sample size for measuring seed yield and growth characteristics? 100? The sample size should be explicitly specified in the methodology.
10) Page 5, Line 132-136, “Table 2. Averages, ranges, and coefficient of variation (CV) for acorn production of the populations...”. The statistical analysis should be conducted to examine the disparity in mean values between GAP and PAP. It is crucial to clearly demonstrate and provide detailed descriptions of any significant differences between the two in the results section. In case there is no significant difference between GAP and PAP, it will directly impact the conclusion drawn from this study.
11) Page 5, Line 137-157, about discussion. The discussion should be grounded in your own findings; however, it appears that your discussion lacks a clear correlation with your own results.
12) Page 6, Line 167-177, “Results of the analysis of variance revealed highly significant (p<0.01), ... p<0.05, ...”. The presentation and description of statistical parameters should be improved to enhance clarity, with a clear indication of significant differences between the two analyzed objects.
13) Page 6, Line 178-181, “Table 4. Relations among the characteristics in two populations: GAP (good acorn production population, above the diagonal) and PAP (poor acorn production population, below the diagonal).”. Were the correlation analyses conducted by pooling GAP and PAP data? The discrepancy between the two datasets is disregarded, and it is necessary to consider and elucidate whether there exists a disparity in separate analysis.
14) Page 8, Line 207-209, “The cumulative contribution of trees to the overall fertility estimates for the acorn productions and each population, and pooled populations were shown in Figure 4. The curve (Figure 4) could be a guide in balancing of contributions for managers such as selection of individuals.” The results lack the necessary interpretation and analysis, and it is essential to interpret these curves in order to better comprehend their meaning and significance.
15) Page 10, Line 255-269, “In conclusion, this study...”. The conclusion needs to be better summarized based on the research results, thus necessitating a revision of the conclusion. This revision should particularly consider the research objectives stated in the introduction, and assess whether these objectives have been accomplished.
Comments on the Quality of English LanguageThe language requires some refinement.
Author Response
Dear Editorial Office of Forests and Reviewer,
Thank you very much for taking the time to review this manuscript (ID: forests- 2630112). Please find the detailed responses below and the corresponding revisions highlighted in track changes in the re-submitted files.
Comment 1: Page 1, Line 29, about introduction, The focus of this study is seed yield; however, the introduction does not address certain pertinent studies on seed yield.
Response 1: We agree with the reviewer. New sentences were added to second paragraph of Introduction by new (Saatcioglu 1978, Hedrick 1919, Vinha et al. 2016) and cited references. However, limited studies were accessible in seed technology and yield of the species.
Comment 2: Page 2, Line 74-75, “A population with good acorn production (GAP, lat. 37°40'555'' N, long. 30°31'941'' E, 930 m asl.) and another population with poor acorn production...”. What are the definitions and criteria for GAP and PAP? Is the current year's yield or the long-term acorn production used as a reference?
Response 2: We agree with the reviewer. However, there is no scale or numerical criteria for GAP and PAP. It was corrected as data collection year, and emphasized as higher and lower in revised version.
Comment 3: Page 2, Line 76-77, “based on criteria”. What is the criteria? The standard should specify specific values or ranges instead of using descriptive terms such as larger diameter, greater height, and straighter stem.
Response 3: We agree with the reviewer. The sentences were corrected. There is no specific value. It is phenotypical.
Comment 4: Page 3, Line 79-80, “The sampled trees were renewed each year at the same sites”, What is the meaning of this sentence? Specifically, what does “renewed” ? Changing sample trees annually according to seed production? If so, how was the three-year experiment conducted?
Response 4: We agree with the reviewer. The mean was tried to clarify.
Comment 5: Page 3, Line 82-83, “Climatic characteristics...”,The location of this climatological site is not specified. The distance between the GAP and PAP sites has not been provided. The presentation of climate characteristics differences between two sites should be considered. Based on your map, it can be inferred that GAP site is situated in closer proximity to the Black Sea, whereas PAP site appears to be more inland. Does the variation in microclimate, particularly humidity account for the disparity in seed yield? Therefore, it is necessary to consider the impact of climate on the outcomes of this study.
Response 5: Details of meteorological station, and distances were given in revised version. We agree with the reviewer. However, the present study did not focus on impact of climate on the outcomes because of limited data.
Comment 6: Page 3, Line 85-86, “Tree height (H), diameter at base (D0), diameter at breast height (D1.30) and crown 85 diameter (CD)”, The process of measuring these indicators should be described in detail, including the specific method used, the tools employed, and the number of samples taken.
Response 6: Details of measurements were given based on the comment in revised version.
Comment 7: Page 3, Line 90-93, “...AN2 in 2022, and AN1 and AN0 in 2023 could represent the acorn production of 2022, 2023, and 2024, respectively, in the future.”. Does it accurately reflect the potential of future production? The fact is that a significant proportion of pollinated seeds fail to reach maturity and achieve seed yield due to various factors, including drought.
Response 7: The sentence was softened based on the comment.
Comment 8: Page 4, Line 96-99, “The one-way analysis of variance (ANOVA) was performed to compare the populations in terms of acorn production and growth characteristics. Phenotypic Pearson correlations among acorn production and growth characteristics were estimated within the populations by SAS package”. The description of the data analysis is rather rudimentary, lacking clear elucidation on the analysis process and parameters. Additionally, the depiction of the analysis results is also quite ambiguous in the result, rendering it difficult to comprehend easily.
Response 8: The model of ANOVA was added in revised version. The correlation analysis is standard.
Comment 9: Page 5, Line 120, “3.1. Acorn production, growth characteristics and their relations”. What was the sample size for measuring seed yield and growth characteristics? 100? The sample size should be explicitly specified in the methodology.
Response 9: “One hundred trees” was changed to “100 trees” in line 77 of subtitle 2.1 to be noticed easily for readers.
Comment 10: Page 5, Line 132-136, “Table 2. Averages, ranges, and coefficient of variation (CV) for acorn production of the populations...”. The statistical analysis should be conducted to examine the disparity in mean values between GAP and PAP. It is crucial to clearly demonstrate and provide detailed descriptions of any significant differences between the two in the results section. In case there is no significant difference between GAP and PAP, it will directly impact the conclusion drawn from this study.
Response 10: We agree with the reviewer. Results of analysis of variance were given for acorn productions at better place in revised version.
Comment 11: Page 5, Line 137-157, about discussion. The discussion should be grounded in your own findings; however, it appears that your discussion lacks a clear correlation with your own results.
Response 11: We agree with the reviewer. A connector sentence from the result was added for discussion.
Comment 12: Page 6, Line 167-177, “Results of the analysis of variance revealed highly significant (p<0.01), ... p<0.05, ...”. The presentation and description of statistical parameters should be improved to enhance clarity, with a clear indication of significant differences between the two analyzed objects.
Response 12: We agree with the reviewer. It was clarified in revised version.
Comment 13: Page 6, Line 178-181, “Table 4. Relations among the characteristics in two populations: GAP (good acorn production population, above the diagonal) and PAP (poor acorn production population, below the diagonal).”. Were the correlation analyses conducted by pooling GAP and PAP data? The discrepancy between the two datasets is disregarded, and it is necessary to consider and elucidate whether there exists a disparity in separate analysis.
Response 13: Impact of the growth characteristics on acorn productions in pooled populations were added based on results of correlation analysis by a sentence.
Comment 14: Page 8, Line 207-209, “The cumulative contribution of trees to the overall fertility estimates for the acorn productions and each population, and pooled populations were shown in Figure 4. The curve (Figure 4) could be a guide in balancing of contributions for managers such as selection of individuals.” The results lack the necessary interpretation and analysis, and it is essential to interpret these curves in order to better comprehend their meaning and significance.
Response 14: We agree with the reviewer. The curves were interpreted in revised version.
Comment 15: Page 10, Line 255-269, “In conclusion, this study...”. The conclusion needs to be better summarized based on the research results, thus necessitating a revision of the conclusion. This revision should particularly consider the research objectives stated in the introduction, and assess whether these objectives have been accomplished.
Response 15: The paragraphs were improved based on the comment.
Comment 16: Response to Comments on the Quality of English Language
Response 16: The authors have the manuscript checked by a colleague fluent in English writing.
We thank to the reviewer for their valuable comments in improving the manuscript.
Sincerely,
Kyu-Suk Kang, Professor
Department of Agriculture, Forestry and Bioresources
College of Agriculture and Life Sciences
Seoul National University, Seoul 08826
Republic of Korea
Round 2
Reviewer 3 Report
Comments and Suggestions for Authors
No other comments, and my concerns have been revised
Author Response
2023-11-02
Dear Editorial Office of Forests and Reviewer,
Thank you very much for taking the time to review this manuscript (ID: forests- 2630112).
As the reviewer 3 did not provide additional comments, we have focused our revisions on other sections of the manuscript.
Please find the detailed responses below and the corresponding revisions highlighted in track changes in the re-submitted files.
We thank to the reviewer for their valuable comments in improving the manuscript.
Sincerely,
Kyu-Suk Kang, Professor
Department of Agriculture, Forestry and Bioresources
College of Agriculture and Life Sciences
Seoul National University, Seoul 08826
Republic of Korea